

# An integrative taxonomic approach reveals *Octopus insularis* as the dominant species in the Veracruz Reef System (southwestern Gulf of Mexico)

Roberto González-Gómez[1,2], Irene de los Angeles Barriga-Sosa[3], Ricardo Pliego-Cárdenas[4], Lourdes Jiménez-Badillo[2], Unai Markaida[5], César Meiners-Mandujano[2] and Piedad S. Morillo-Velarde[6]

[1] Posgrado en Ecología y Pesquerías, Universidad Veracruzana, Boca del Río, Veracruz, México
[2] Instituto de Ciencias Marinas y Pesquerías, Universidad Veracruzana, Boca del Río, Veracruz, México
[3] Departamento de Hidrobiología, Universidad Autónoma Metropolitana-Iztapalapa, Ciudad de México, México
[4] División de Estudios Profesionales, Facultad de Ciencias, Universidad Nacional Autónoma de México, Ciudad de México, México
[5] Laboratorio de Pesquerías Artesanales, El Colegio de la Frontera Sur (CONACyT), Lerma, Campeche, México
[6] CONACyT- Instituto de Ciencias Marinas y Pesquerías, Universidad Veracruzana, Boca del Río, Veracruz, México

Corresponding author
César Meiners-Mandujano,
cmeiners@uv.mx

## ABSTRACT

The common octopus of the Veracruz Reef System (VRS, southwestern Gulf of Mexico) has historically been considered as *Octopus vulgaris*, and yet, to date, no study including both morphological and genetic data has tested that assumption. To assess this matter, 52 octopuses were sampled in different reefs within the VRS to determine the taxonomic identity of this commercially valuable species using an integrative taxonomic approach through both morphological and genetic analyses. Morphological and genetic data confirmed that the common octopus of the VRS is not *O. vulgaris* and determined that it is, in fact, the recently described *O. insularis*. Morphological measurements, counts, indices, and other characteristics such as specific colour patterns, closely matched what had been reported for *O. insularis* in Brazil. In addition, sequences from cytochrome oxidase I (COI) and 16S ribosomal RNA (r16S) mitochondrial genes confirmed that the common octopus from the VRS is in the same highly supported clade as *O. insularis* from Brazil. Genetic distances of both mitochondrial genes as well as of cytochrome oxidase subunit III (COIII) and novel nuclear rhodopsin sequences for the species, also confirmed this finding (0–0.8%). We discuss our findings in the light of the recent reports of octopus species misidentifications involving the members of the 'O. vulgaris species complex' and underscore the need for more morphological studies regarding this group to properly address the management of these commercially valuable and similar taxa.

## INTRODUCTION

Many octopus fisheries are of high economic local importance (*Jiménez-Badillo, 2010*; *Rosas et al., 2014*). Despite this fact, in many cases, the taxonomic identity of the targeted species remains unknown or has been long taken for granted because official fishery statistics do not attempt to distinguish different species (*Domínguez-Contreras et al., 2018*). Food and Agriculture Organization (FAO) catch statistics currently include only four octopus species names, *Octopus vulgaris* Cuvier 1797, *O. maya* Voss & Solís-Ramírez, 1966, *Eledone cirrhosa* (Lamarck, 1798) and *E. moschata* (Lamarck, 1798), with the rest being classified as unidentified octopuses (*Norman, Finn & Hochberg, 2016*). However, as many finfish stocks are collapsing worldwide, commercial interests are shifting towards the exploitation of cephalopod resources. Therefore, as the value of octopus fisheries continues to increase, the need for rigorous taxonomic knowledge is greater than ever before (*Norman & Hochberg, 2005*). This is particularly important in Mexico because it is the largest American octopus producer (*Norman & Finn, 2016*).

The difficulty of correctly assigning the taxonomic identity of octopus species partially lies in the existence of several species complexes comprising taxa that share superficial morphological similarity (*Norman, 1992*; *Roper, Gutierrez & Vecchione, 2015*; *Amor et al., 2016*; *Gleadall, 2016*) and that are currently treated under the catch-all species names 'vulgaris', 'macropus' and 'defilippi' (*Norman & Hochberg, 2005*). Moreover, the genus *Octopus* has been used, to date, to include the vast majority of described shallow-water octopuses, including taxa designated as 'unplaced' (*Norman & Hochberg, 2005*; *Norman, Finn & Hochberg, 2016*). However, recent molecular studies have proven that the genus *Octopus* is polyphyletic and contains a number of distinct and divergent clades (*Guzik, Norman & Crozier, 2005*; *Acosta-Jofré et al., 2012*). In this paper, we refer to the genus *Octopus* as the group of species including the 'Octopus vulgaris species complex' and its close relatives, *sensu Norman, Finn & Hochberg (2016)*. The 'Octopus vulgaris species complex' currently comprises the type species of the group, *O. vulgaris sensu stricto* (*s. s.*), found in the Mediterranean Sea, and the central and north-east Atlantic Ocean, plus four more 'types' inhabiting different geographical areas; type I (tropical western central Atlantic Ocean), type II (subtropical south-west Atlantic Ocean), type III (temperate South Africa and the southern Indian ocean) and type IV (subtropical/temperate east Asia) (*Amor et al., 2016*; *Norman, Finn & Hochberg, 2016*). The representative of the complex in Mexican Atlantic waters, *O.* 'vulgaris' type I, is of high fisheries value, with annual catches averaging almost 7,000 *t* for the last 10 years (*National Aquaculture and Fishing Commission of Mexico (CONAPESCA), 2018*).

Despite the similarities of this closely related taxa, in recent years, more detailed and consistent diagnoses and descriptions have described new octopus species, for example *O. insularis* Leite & Haimovici, 2008 and *O. tayrona* Guerrero-Kommritz & Camelo-Guarin, 2016; as a consequence, it is now known that the *O.* 'vulgaris' type I is a group that comprises several species. Most species in this complex have yet to be distinguished using morphological and meristic characters (*Amor et al., 2016*). A recent assessment in different coastal and oceanic regions along the Tropical Northwestern

Atlantic and Tropical Southwestern Atlantic revealed that several commercially fished octopus specimens previously identified as *O. vulgaris* were being mislabeled and were in fact either *O. maya* or *O. insularis*, thus proving the common misidentification that often occurs among the exploited octopus species in the area (*Lima et al., 2017*). Proper identification of organisms is necessary to monitor biodiversity at any level (*Vecchione & Collette, 1996*) and it is particularly important in the case of commercially exploited species because it allows the effective management of their stocks by considering specific biological features and thus defining particular conservation proposals to prevent overexploitation (*Ward, 2000*; *Lima et al., 2017*).

Misidentification among the species of the genus *Octopus* has been attributed to a general external resemblance as well as to similar skin texture and colour patterns (*Norman & Hochberg, 2005*). However, despite the superficial morphological similarity among the species conforming the 'Octopus vulgaris species complex', recent studies have demonstrated that closely related species can be identified based on discrete phenotypic differences (*Huffard & Hochberg, 2005*; *Leite et al., 2008*; *Gleadall, 2016*). Recently, *Amor et al. (2016)* carried out the most comprehensive morphological and molecular-based assessment of the *O. vulgaris* species complex to date and found that all members of the group could be distinguished based on morphological analyses in which male morphology, (e.g. sexual traits) proved to be a more reliable indicator of species-level relationships in comparison with female morphology. As noted by *Pomiankowski & Moller (1995)*, sexual traits (e.g. the hectocotylus), are usually more phenotypically variable than non-sexual traits among close relatives, making them ideal characters to distinguish between species (*Amor et al., 2016*).

In the southwestern Gulf of Mexico, the important shallow-water octopus fishery operating in the Veracruz Reef System (VRS) has historically been attributed to *O. vulgaris* (*Jiménez-Badillo & Castro-Gaspar, 2007*; *Méndez-Aguilar, Jiménez-Badillo & Arenas-Fuentes, 2007*; *Jiménez-Badillo et al., 2008*). However, *Flores-Valle et al. (2018)* determined the occurrence of *O. insularis* in the VRS and suggested that the common octopus of this system might not be *O. vulgaris* but *O. insularis* instead, originally described in Brazil. The pitfalls associated with a single approach when trying to assign the status of a certain taxon can be avoided by using an integrative taxonomic approach, which aims to delimit the units of life's diversity from multiple and complementary perspectives (*Dayrat, 2005*). Thus, this approach overcomes biases associated to individual lines of evidence, increasing the information on which taxonomic hypotheses are tested (*Chesters et al., 2012*). In accordance, the aim of this study was to make a comprehensive description of the VRS common octopus following an integrative taxonomic approach to clarify its taxonomic status by means of both morphological and genetic analyses, including sequences from three mitochondrial (cytochrome oxidase I (COI), cytochrome oxidase subunit III (COIII), 16S ribosomal RNA (r16S)), and one nuclear region, rhodopsin.

# MATERIAL AND METHODS

## Collection sites

The study area lies within the VRS National Park, which is located in the southwestern region of the Gulf of Mexico, off the coast of Veracruz, between 19.04–19.26°N and
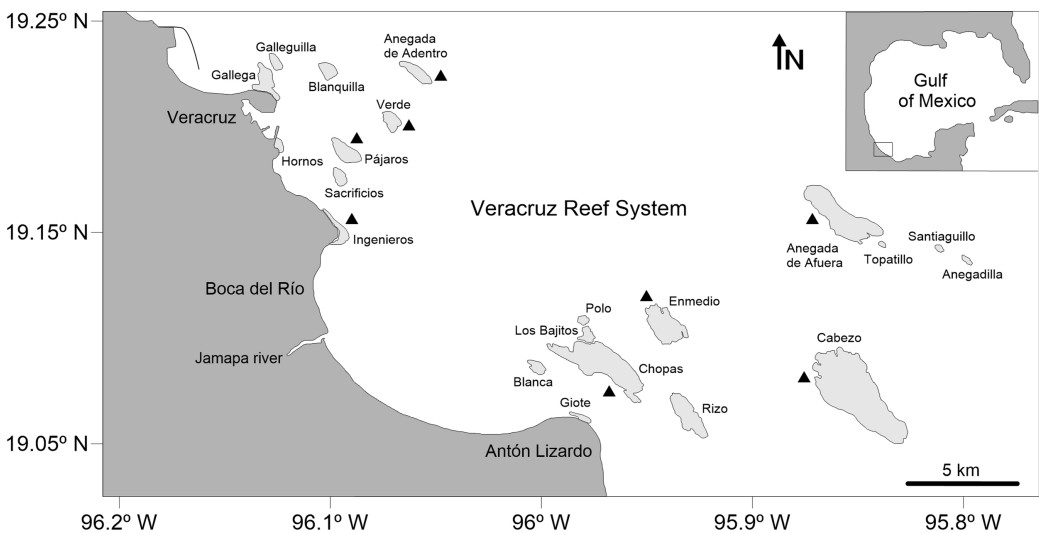

**Figure 1 Map of the Veracruz Reef System, southwestern Gulf of Mexico.** Black triangles indicate collecting sites (specimens were collected in the reef lagoon and fore-reef). Degrees are in decimal notation.

95.77–96.20°W and includes 28 reefs and six cayes and islands in an area of 65,516 ha (*Diario Oficial de la Federación (DOF), 2012*, *2017*). A total of 52 octopuses were randomly selected from the commercial catches of the artisanal fishery between May and November 2017. All specimens were collected by hand or using a hook while snorkeling in shallow waters (up to three m) of the reef lagoon and adjacent areas of eight reefs within the VRS: Enmedio, Anegada de Afuera, Anegada de Adentro, Cabezo, Chopas, Verde, Pájaros and Ingenieros (Fig. 1). These reefs were selected to have a good sampling representation of both northern and southern reef subsystems, located off the city of Veracruz and the village of Antón Lizardo, respectively, and divided by the outlet of the Jamapa River (*Horta-Puga, 2003*). Oceanographic characteristics in the reefs are given by the Gulf Common Water, with a mean salinity of 36.5 practical salinity unit (PSU) and temperatures between 21.2 and 30.0 °C (*Mateos-Jasso et al., 2012*). The benthic habitat in the sampling sites is characterized by the presence of numerous scattered patches of seagrass, sand, coral rubble, several species of algae, isolated branching and massive corals and an underlying rocky basement constituted by remains of *Porites porites* (Pallas, 1766) mixed with *Siderastrea radians* (Pallas, 1766) and *Pseudodiploria clivosa* (Ellis & Sollander, 1786) (*Chávez, Tunnell & Withers, 2007*).

## Morphological study

Octopus specimens used for morphological analysis were ice-stored in zip-lock plastic bags for 48 h, then fixed in 10% formalin and finally preserved in 70% ethanol after rinsing in running tap water. The measurements and indices used for the description follow *Roper & Voss (1983)* and *Norman, Hochberg & Lu (1997)* with the exception of the arm length index (ALI), which is defined here as length of the longest arm as a percentage of the total length (TL) (not mantle length) and sucker counts, which included
all suckers of intact arms instead of only those in the basal half of the arms. Reproductive terminology follows *Huffard & Hochberg (2005)*. Web depth values of sectors B, C and D are the mean value of right and left sides.

In all, 52 octopuses were analysed and their morphological characters recorded. However, morphological and meristic data presented in the results section were based on submature and mature specimens only (e.g. maturity stages II–IV, $n = 18$, *Leite et al., 2008*; *Guerra et al., 2010*), because counts and relative measurements in immature specimens undergo considerable change in early growth stages and can cause overlap in otherwise valid diagnostic characters (*Norman, Hochberg & Lu, 1997*).

The small structures, such as ligula, calamus, radula, spermatophores and eggs were measured with the aid of an ocular micrometre in a binocular microscope (Zeiss Stemi 2000-C; ZEISS Group, Oberkochen, Germany). All measurements are in millimetres and the weights in grams unless stated otherwise.

The sex of the specimens was assigned by the observation of the reproductive organs and the stage of maturity classified as: I (Immature), II (Maturing), III (Mature) and IV (Post-maturation) following the macroscopic scale for stages of gonadal maturity proposed by *Lima et al. (2014)*.

Digestive tracts and reproductive organs were dissected in some specimens for examination and description. Illustrations were edited with Adobe Photoshop CS6 from high-resolution photographs taken with a digital camera (Nikon D90; NIKON CORPORATION, Tokyo, Japan). Beaks and radula were photographed after cleaning with a saturated solution of sodium hydroxide (NaOH).

## Statistical analyses of morphological data

Preliminary observations suggested the existence of morphological differences between the VRS common octopus and *O. vulgaris s. s.* To further investigate these differences, we performed multivariate analyses with PRIMER 7 v7.0.13 (PRIMER-E Ltd; Plymouth, United Kingdom) comparing recorded morphological data on the VRS common octopus with published data on *O. insularis* and *O. vulgaris s. s.* (included in Table S2 from *Amor et al., 2016*). In all, 10 morphological traits were compiled in a matrix including information of the three taxa; these were: head width index (HWI), calamus length index (CLI), ligula length index (LLI), enlarged sucker diameter index (eSDI), hectocotylized arm index (HcAI), hectocotylized arm sucker count (HASC), funnel length index (FLI), mantle width index (MWI), web depth index (WDI) and terminal organ length index (TOLI). Analysis of morphological traits was limited to male specimens to maximize the number of indices and counts used, minding that male morphology has proven a more reliable indicator at a species-level compared to female morphology (*Amor et al., 2016*). Given that measurements of some traits could not be obtained for particular individuals because of damage, all missing data were replaced with the mean of that trait for each taxa, as missing data are not permitted in the analysis (*Allcock, Strugnell & Johnson, 2008*; *Amor et al., 2016*). Morphological traits were transformed to zero mean and unit standard deviation, thus allowing for comparisons of traits despite having different measurement scales (*Allcock, Strugnell & Johnson, 2008*; *Amor et al., 2016*). A resemblance matrix

based on Euclidean distance was calculated for the normalized traits and a principal coordinate ordination (PCO) plot, with vector overlay using Pearson correlation >0.6, was used to visualize the natural groupings of the samples (*Roura et al., 2016*). The statistical significance of the differences observed in the PCO plot was further tested with a one-way analysis of similarity (ANOSIM) (*Allcock, Strugnell & Johnson, 2008*). This test gives an *R*-value indicative of the difference between samples as well as a *p*-value for the significance of that difference. *R*-values close to 1 indicate large differences among samples while values closer to 0 indicate lesser differences (*Clarke & Warwick, 2001*). The similarity percentage analysis (SIMPER, *Clarke, 1993*) was used to determine the percentage contribution of each morphological trait to the average square distance between the compared taxa. Results from analyses were deemed significant at $p < 0.05$.

## Genetic identification and relationships of octopus specimens

To perform the genetic identification of the VRS common octopus, muscle tissue samples from 20 octopuses were preserved in non-denatured 95% ethanol following the procedure suggested by *Wall, Campo & Wetzer (2014)* and maintained at −4 °C for 72 h for tissue fixation before processing for DNA extraction. All specimens used for genetic identification were also morphologically analysed, to strengthen conclusions drawn within an integrative taxonomic approach.

Total DNA was extracted from arm tissue using the Wizard® Genomic DNA Purification kit (Promega®, Madison, WI, USA). PCR amplifications for mitochondrial COI, COIII and r16S genes and the rhodopsin nuclear marker were carried out using QIAGEN® Kit PCR reagent system (Valencia, CA, USA). Each 25 μL reaction contained 1.0 μL of $MgCl_2$ (2.0 mM), 10 μM each primer, 200 μM each dNTP, 2.5X PCR Buffer and 2.5U *Taq* Polymerase. Primers for COI were those described by *Allcock, Strugnell & Johnson (2008)*, the COIII ones were from *Barriga-Sosa et al. (1995)*, the r16S ones were those from *Simon, Franke & Martin (1991)* and Rhodopsin primers were from *Strugnell (2004)*. PCR reactions were conducted in a Mycycler (Bio-Rad®) thermocycler using the annealing temperatures of 50 °C for rhodopsin and 49 °C for COI, 52 °C for r16S, and 32 °C for COIII under the following conditions: an initial cycle of denaturing at 94 °C for 2 min; followed by 30 cycles at 94 °C for 45 s, an annealing step for 60 s, and extension step at 72 °C for 90 s, and finally an extension cycle at 72 °C for 5 min.

Sequencing reactions on both directions were carried out using Macrogen (Seoul, South Korea) services. Additional sequences of several octopod species were obtained from GenBank for comparison. The alignments of the sequences were verified with the respective translation of amino acids for COI, COIII and rhodopsin. Genetic distances were calculated for each gene region using the Tamura–Nei model (*Tamura & Nei, 1993*). Bootstrap support was estimated using 500 iterations. All these analyses were implemented in Mega 7.0 (*Kumar, Stecher & Tamura, 2016*).

JModelTest (*Darriba et al., 2012*) was used to select the best evolutionary model for each gene region. The appropriate model was chosen based on 'goodness of fit' via the Akaike information criterion. The best fit model for COI was GTR+I+G and TIM3+G (topology GTR+G) for r16S. Phylogenetic reconstruction was conducted by using each gene

separately. Bayesian Inference was run using MrBayes 3.1.2. (*Ronquist & Huelsenbeck, 2003*), only for COI and r16S genes, because of limited or absence of homologous sequences in GenBank for *O. insularis* COIII and rhodopsin, respectively. '*Octopus*' *cyanea* Gray, 1849 was selected as outgroup on the basis of its close phylogenetic relation to the internal group (*Amor et al., 2014, 2017*). Analyses were started from random trees, and they were run for five million generations for each data set and sampling the Markov chain every 1,000 generations. The programme Tracer v1.3 (*Rambaut et al., 2014*) was then used to ensure Markov chains had reached stationarity and to determine the correct 'burn-in' for the analysis. The analysis converged after 500,000 generations with effective sample size (ESS) values > 200 for all parameters.

# RESULTS

## Diagnosis of the Veracruz Reef System common octopus

Medium to large sized animals with dorsal mantle length (ML) up to 189 mm and total wet weight (TW) up to 1,811 g; hectocotylized arm bearing 103–146 suckers; small ligula (LLI 0.92–1.65) and relatively long calamus (CLI 40.79–58.56); slightly enlarged suckers in mature males (eSDI 8.87–13.75); 8–11 lamellae on outer demibranch; one large papilla and several smaller ones over each eye. Live animals creamish in colour, showing a distinct red/white reticule in the inner of arms when hidden in the den and still visible in freshly dead specimens. No ocellus present.

## Morphological description of the Veracruz Reef System common octopus

The following description is based on 14 males and four females, all of them in maturity stages II–IV. Most relevant counts, measurements and indices are given in Tables 1 and 2 and in Table S1.

Medium to large-sized organisms (up to 696 mm TL and 1,811 g TW) with muscular body (Fig. 2A). Mantle wide (max 189 mm ML) and saccular. Head wide (HWI 27.67–50.13) and pallial aperture moderately wide (pallial aperture index (PAI) 33.45–61.66). Funnel tubular (FLI 26.79–49.47) with funnel organ well defined and W shaped (Fig. 2B). Most common arm formula is: IV>II>III>I (right) and IV>III>I>II (left). Arms are wide (arm width index 12.82–23.15) and relatively short (ALI 77.53–87.22). Third right arm in males hectocotylized, bearing 103–146 suckers and normally shorter than opposite one (opposite arm index 77.06–90.75). Spermatophoric groove well defined, running ventrally along the arm and ending at a relatively big calamus (CLI 40.79–58.56). Ligula small (LLI 0.92–1.65) (Fig. 2C). Suckers in normal arms between 103 and 267 (normal sucker diameter index 7.05–10.42). Mature males have enlarged suckers in arms II and III, normally between rows 13 and 16, more conspicuous in large-sized specimens (eSDI 8.87–13.75) (Fig. 2E). Females do not have them. Stylets present, wide and hockey club-shaped (Fig. 2D). Web moderately deep (WDI 16.35–24.91), typical web formula D>C>E>B>A. Gills with 8–11 lamellae per outer demibranch.

Digestive system consisting of a big buccal mass with conspicuous anterior salivary glands, narrow oesophagus, big triangular posterior salivary glands, slender crop, wide

**Table 1  Morphological measurements and counts of the VRS common octopus.**

| Parameter | Males (n = 14) | | | Females (n = 4) | | |
|---|---|---|---|---|---|---|
| | Min | Mean | Max | Min | Mean | Max |
| Total weight | 113 | 850.2 | 1811 | 595 | 1014.0 | 1326 |
| Total length | 375 | 504.8 | 696 | 515 | 564.3 | 630 |
| Mantle length (dorsal) | 101 | 130.4 | 189 | 113 | 137.8 | 157 |
| Mantle width | 53.2 | 75.9 | 110.0 | 77.6 | 86.1 | 90.8 |
| Head width | 33.1 | 46.1 | 65.3 | 32.6 | 47.7 | 61.9 |
| Ligula length | 2.6 | 4.2 | 5.8 | – | – | – |
| Calamus length | 1.1 | 2.1 | 3.2 | – | – | – |
| Hectocotylized arm sucker count | 103 | 122 | 146 | – | – | – |
| Normal sucker diameter | 7.2 | 10.9 | 16.7 | 9.5 | 11.1 | 12.2 |
| Enlarged sucker diameter | 9.1 | 14.1 | 19.8 | – | – | – |
| Terminal organ length | 10.8 | 14.1 | 18.0 | – | – | – |
| Arm Length | | | | | | |
| 1 | 243/292 | 345/361.5 | 421/488 | 280/278 | 336.3/379.5 | 416/481 |
| 2 | 242/253 | 384.7/354.8 | 527/501 | 389/175 | 453.7/360.5 | 536/484 |
| 3 | 257/293 | 342.1/351.6 | 446/388 | 281/412 | 363/425.5 | 445/439 |
| 4 | 284/256 | 388.1/378.1 | 590/539 | 406/275 | 456/413 | 512/502 |
| Arm sucker count | | | | | | |
| 1 | 162/178 | 192.5/201 | 219/228 | 103/125 | 167.6/182.5 | 235/240 |
| 2 | 162/170 | 201.7/195.6 | 227/222 | 158/124 | 215.5/192.3 | 249/232 |
| 3 | 103/145 | 122/190 | 146/225 | 148/200 | 161.5/201 | 175/202 |
| 4 | 113/175 | 211.3/214.4 | 257/267 | 208/160 | 225.5/209.3 | 263/249 |
| Arm width | 15.1 | 21.2 | 28.1 | 16.8 | 20.7 | 25.3 |
| Web depth | | | | | | |
| A | 37.8 | 53.7 | 74.9 | 36.7 | 51.8 | 64.3 |
| B | 52.3 | 69.6 | 96.4 | 55.8 | 74.1 | 86.4 |
| C | 62.9 | 85.7 | 118.6 | 88.1 | 92.5 | 99.0 |
| D | 62.4 | 86.7 | 128.0 | 86.2 | 92.6 | 98.4 |
| E | 40.6 | 69.9 | 104.0 | 70.7 | 74.8 | 84.5 |
| Funnel length | 30.8 | 41.0 | 55.5 | 36.4 | 44.4 | 50.1 |
| Eye lens diameter | 5.2 | 7.3 | 10.1 | 5.4 | 7.6 | 8.8 |
| Spermatophore length | 33.8 | 46.4 | 57.2 | – | – | – |
| Spermatophore width | 0.6 | 0.7 | 0.9 | – | – | – |
| Pallial aperture | 35.3 | 52.3 | 77.5 | 57.9 | 62.4 | 69.8 |
| Gill count | 8 | 9.6 | 11 | 9 | 9.8 | 10 |

stomach and spiral caecum with three whorls (Fig. 3A). The ink sac is embedded in the digestive gland. Intestine long, muscular. Anal flaps present. The beaks are strong, with prominent rostrum and wide wings (Figs. 3C and 3D). Radula with seven teeth and two marginal plates per transverse row. Rachidean tooth with one lateral cusp at each side and symmetric seriation every three teeth ($A_3$) (Figs. 3B and 3E).

Table 2 Morphological indices of the VRS common octopus.

| Parameter | Males (n = 14) | | | Females (n = 4) | | |
|---|---|---|---|---|---|---|
| | Min | Mean | Max | Min | Mean | Max |
| Head width index | 27.67 | 35.80 | 50.13 | 28.80 | 34.33 | 43.90 |
| Mantle width index | 45.85 | 58.14 | 74.24 | 55.43 | 63.49 | 80.21 |
| Ligula length index | 0.92 | 1.25 | 1.65 | – | – | – |
| Calamus length index | 40.79 | 49.18 | 58.56 | – | – | – |
| Normal sucker diameter index | 7.05 | 8.32 | 10.42 | 7.43 | 8.04 | 8.58 |
| Enlarged sucker diameter index | 8.87 | 10.64 | 13.75 | – | – | – |
| Mantle arm index | 26.60 | 31.10 | 35.02 | 24.34 | 28.78 | 31.75 |
| Arm length index | 77.53 | 82.73 | 87.22 | 83.30 | 84.91 | 85.64 |
| Opposite arm index | 77.06 | 85.65 | 90.75 | – | – | – |
| Arm width index | 12.82 | 16.40 | 23.15 | 13.44 | 15.06 | 17.94 |
| Hectocotylized arm index | 229.56 | 265.21 | 306.88 | – | – | – |
| Funnel length index | 26.79 | 31.48 | 37.00 | 31.91 | 39.16 | 49.47 |
| Pallial aperture index | 33.45 | 40.07 | 50.48 | 39.04 | 46.26 | 61.66 |
| Eye lens diameter index | 0.04 | 0.06 | 0.07 | 0.04 | 0.06 | 0.08 |
| Web depth index | 16.35 | 21.25 | 24.91 | 18.47 | 19.79 | 21.52 |
| Terminal organ length index | 7.51 | 11.28 | 13.29 | – | – | – |
| Spermatophore length index | 28.06 | 32.43 | 38.68 | – | – | – |

Female reproductive system consisting of a large and round ovary in mature females, with thin oviducts and oviductal glands small and rounded (Fig. 4A). Eggs small (Fig. 4C); mean length and width of immediately spawned eggs were 2.23 ± 0.05 and 0.92 ± 0.06 mm, respectively (mean ± SD). Male reproductive system comprises a large testis followed by a long and thin vas deferens packed in a membranous sac. Spermatophoric gland opens in an atrium with the accessory gland and the spermatophore storage sac (maximum 70 spermatophores). The terminal organ is short and has a rounded diverticulum (Fig. 4B). Spermatophores are medium sized (spermatophore length index (SpLI) 28.06–38.68, Fig. 4D).

In fixed organisms, skin presents well defined polygonal patches with distinct grooves and is covered in papillae in the dorsal surface; ventrally, this occurs to a lesser extent. Colour varies from yellowish to violet dorsally and from cream to grey-brown ventrally. There is one large cirrus and some smaller ones over each eye (Fig. 5A). In live specimens colour varies from pale yellow to reddish-brown, being cream the most common. Among the most distinctive chromatic components observed in live or fresh specimens we could observe: dark/light bars alternating around the eye, a red/white reticulate pattern in the ventral part of the arms when the animal was hidden in the den, and a blue–green circle around the eye (Fig. 5B).

## Morphological analysis

Multivariate combinations of morphological traits were successful in distinguishing among the three taxa compared. In the PCO plot, the first two components explained

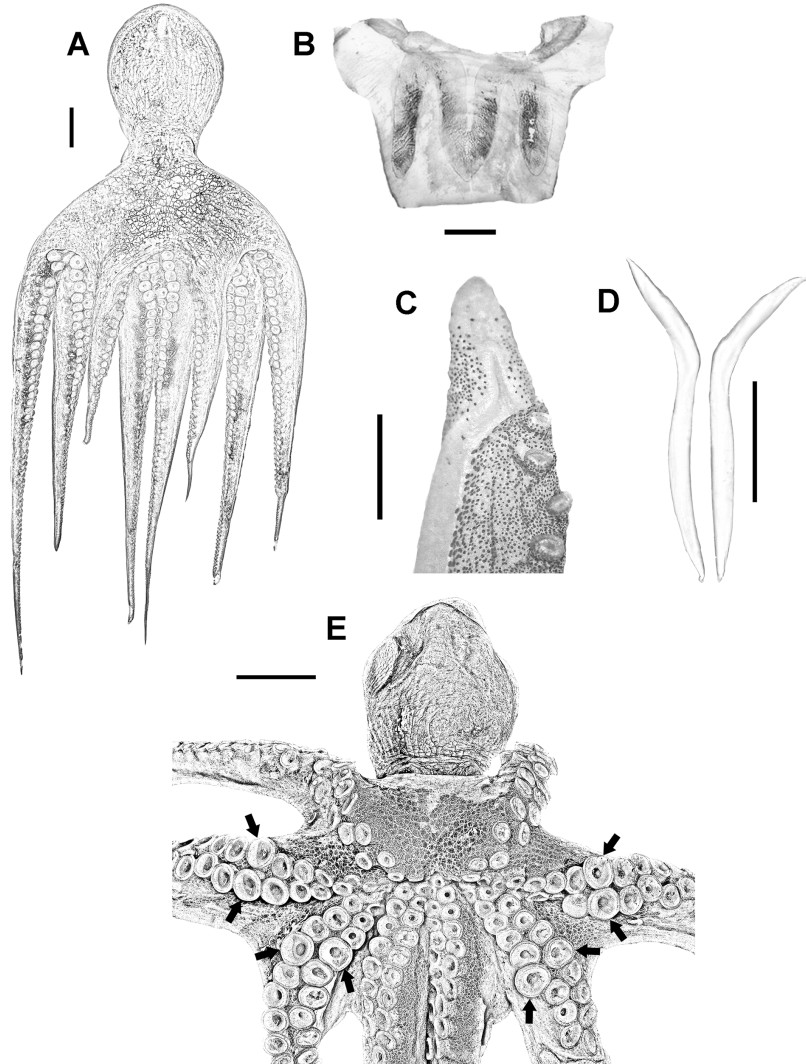

**Figure 2 Veracruz Reef System common octopus.** (A) Dorsal view of a 164 mm ML male. (B) W-shaped funnel organ of a 122 mm ML male. (C) Ligula and calamus of a 101 mm ML male. (D) Pair of stylets of a 124 mm ML male. (E) Ventral view of a male specimen showing the position of enlarged suckers in arms II and III of mature males. Scale bars: A, five cm; B, one cm; C, two mm; D, one cm; E, two cm.

72% of the total variance (Fig. 6). The first component explained 57.8% of the variance and was strongly correlated with eSDI (loading: 0.906), TOLI (0.888), HWI (0.871) and HASC (0.863). The second component explained 14.2% of the variance and was correlated mainly with CLI (0.807) and LLI (0.623). The PCO plot showed a complete differentiation between the VRS common octopus and *O. vulgaris s. s.*, mainly along the first component of the PCO (Fig. 6) with *O. vulgaris s. s.* showing high PC1 loadings attributed to higher sucker numbers in the hectocotylized arm and relatively larger enlarged suckers. *O. insularis* and the VRS common octopus showed the least discrimination, although the former had relatively higher HWI, WDI and MWI than the latter. ANOSIM test confirmed the significance of the observed differences (Global $R = 0.751$, $p < 0.001$) and

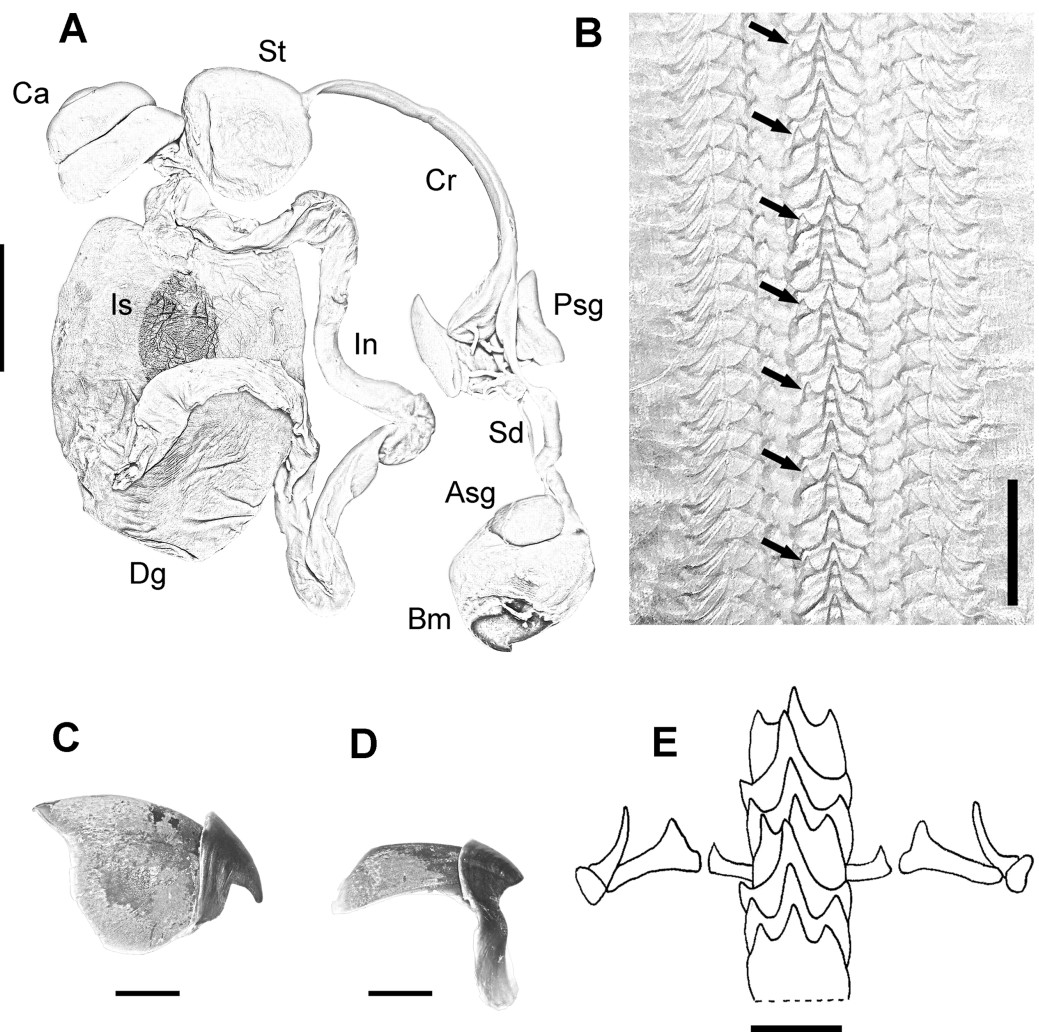

**Figure 3 Digestive system of the VRS common octopus.** (A) Digestive system of a 164 mm ML male. (B) Radula of a 124 mm ML male showing A₃ seriation. (C) Upper beak of a 124 mm ML male. (D) Lower beak of a 124 mm ML male. (E) Radula. Abbreviations: Asg, Anterior salivary glands; Bm, Buccal mass; Ca, Caecum; Cr, Crop; Dg, Digestive gland; In, Intestine; Is, Ink sac; Psg, Posterior salivary glands; Sd, Salivary duct; St, Stomach. Scale bars: A, two cm; B, 50 μm; C, five mm; D, five mm; E, 250 μm.

pairwise comparisons showed the existence of significant differences in morphological traits between all taxa pairs, indicating they were greatest between the VRS common octopus and *O. vulgaris s. s.* ($R = 0.943$, $p < 0.001$), intermediate between *O. vulgaris s. s.* and *O. insularis* from Brazil ($R = 0.664$, $p < 0.001$) and smallest between this latter taxon and the VRS common octopus ($R = 0.66$, $p < 0.001$). SIMPER analysis showed that the main morphological traits responsible for the differences between *O. vulgaris s. s.* and both *O. insularis* from Brazil and the VRS common octopus were reproductive traits (e.g. HASC, TOLI, eSDI). In contrast, the main traits differentiating these last taxa were related to the shape of the web and the mantle: WDI and MWI, respectively, accounting for nearly 40% of the observed differences (Table 3).

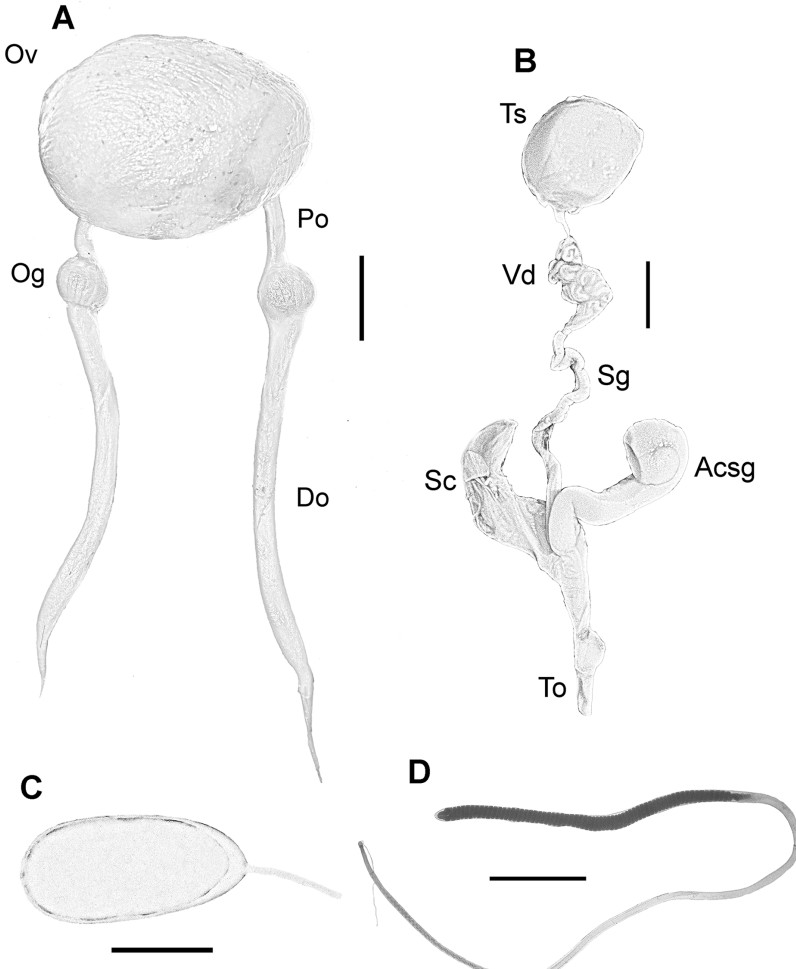

**Figure 4  Reproductive system of the VRS common octopus.** (A) Reproductive system of a 156 mm ML female. (B) Reproductive system of a 159 mm ML male. (C) Egg of a 113 mm ML female. (D) Spermatophore of a 159 mm ML male. Abbreviations: Acsg, Accesory spermatophoric gland; Do, Distal oviduct; Og, Oviductal gland; Ov, Ovary; Po, Proximal oviduct; Sc, Spermatophore storage sac; Sg, Spermatophoric gland; To, Terminal organ; Ts, Testis; Vd, Vas deferens. Scale bars: A, 10 mm; B, 10 mm; C, one mm; D, five mm.                                     

## Genetic identification and relationships of octopus specimens

Sequences from 20 specimens (GenBank accession numbers: MH550422–MH550467) resolved two and three haplotypes for r16S (400 pb, $N = 18$) and COI (605 pb, $N = 18$), respectively. Haplotype 1 for r16S ($N = 17$), was a shared haplotype with *O. insularis* from the northern coast of Brazil (KF843956–7, 60–62, 64–66), whereas Haplotype 2 was a novel one for this study. For COI, two haplotypes were shared with those reported elsewhere. For instance, Haplotype 1 ($N = 13$), was a shared type with *O. insularis* from the coast of Brazil (KX611855, KF844000–1, 5, 7, 9 & 19). Haplotype 2 ($N = 4$) was shared with type KX611857 and also one novel haplotype was identified for the VRS. For COIII, nine specimens from the VRS shared a unique haplotype from GenBank (AJ012123). The only haplotype resolved for the nuclear gene rhodopsin is novel for the

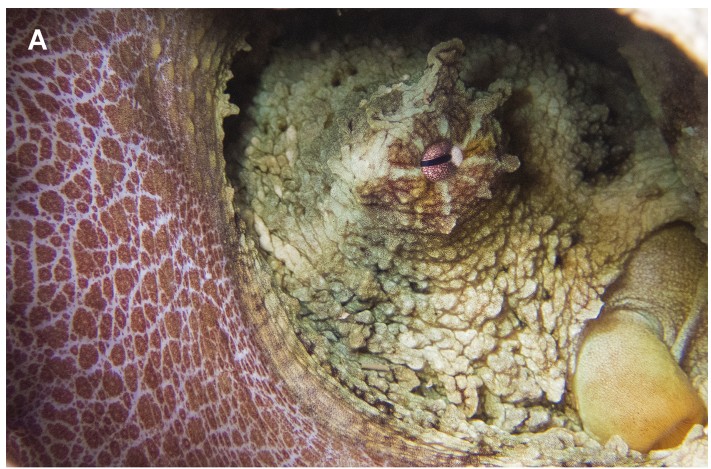

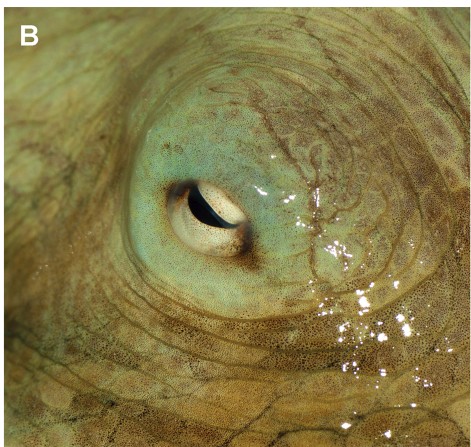

**Figure 5  Skin and colour patterns of the VRS common octopus.** (A) Living specimen hidden in a den showing a characteristic red/white reticulate pattern in the arms and alternating light/dark bars around the eye. One large cirrus and some other small ones can also be observed over the eye. Photograph taken at Enmedio reef, Veracruz. (B) Fresh specimen showing the blue–green colour around the eye (Photo credits A, B: Roberto González-Gómez).               

species (MH550449), since there are no homologous sequences for *O. insularis* in GenBank.

The COI, COIII, r16S and rhodopsin genetic distances between the analysed specimens from the VRS, and *O. insularis* from Brazil resolved from no genetic divergence to very low values between them (0.0–0.6%, see Table 4). One novel rhodopsin haplotype was resolved for the species with genetic distances from 0.5% to 1.7% with respect to the species that conform the American octopus clade (*O. mimus* Gould, 1852, *O. bimaculatus* Verrill, 1883 and *O. bimaculoides* Pickford & McConnaughey, 1949, see Table 4).

The phylogenetic topologies for both mitochondrial regions COI and r16S, recovered two main clades (pp = 0.9), one of them containing species from America (*O. bimaculatus*, *O. bimaculoides*, *O. insularis*, *O. maya* and *O. mimus*) and the other one containing *O. vulgaris* types, *O. sinensis* d'Orbigny, 1834, *O. tetricus* Gould, 1852 and *O. hummelincki* Adam, 1936. All specimens collected in the VRS fell within a highly supported

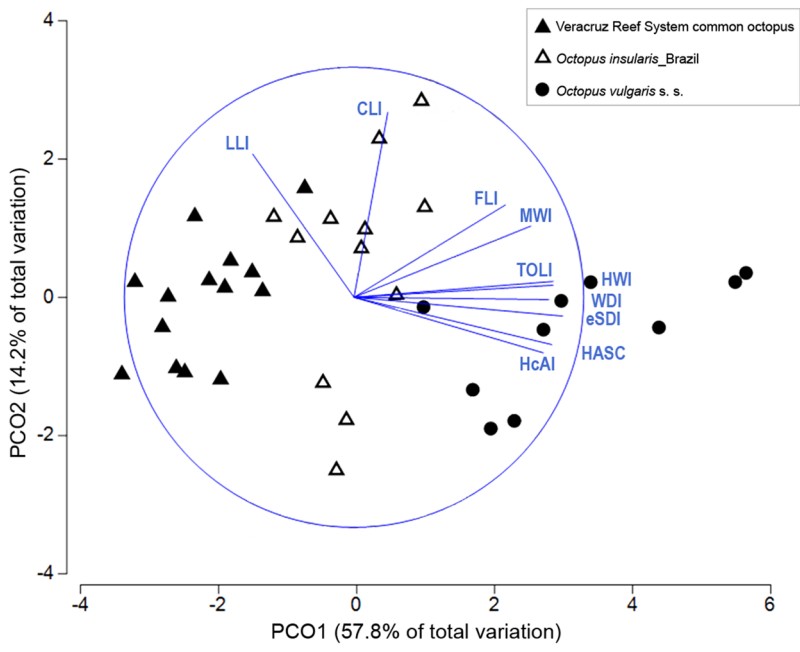

**Figure 6 Principal coordinate ordination (PCO) plot of morphological traits.** Bi-plot showing differences in morphological traits among octopod taxa with vector overlay from Pearson's correlation >0.6. The blue circle represents a maximal vector. Each symbol represents a specimen. HcAI, hectocotylized arm index; HASC, hectocotylized arm sucker count; eSDI, enlarged sucker diameter index; WDI, web depth index; HWI, head width index; TOLI, terminal organ length index; MWI, mantle width index; FLI, funnel length index; CLI, calamus length index; LLI, ligula length index.

monophyletic clade with both gene regions (pp = 1 and pp = 0.88, for COI and r16S, respectively) along with *O. insularis* individuals from Brazil (Figs. 7 and 8).

## DISCUSSION

Our integrative taxonomic study confirms that the VRS common octopus is *O. insularis*. The measurements, counts and indices of octopuses analysed in this study, not previously distinguished from *O. vulgaris*, as well as the shape and size of beaks, stylets, spermatophores, eggs and other features such as specific colour patterns almost entirely match those reported by *Leite et al. (2008)*, *Leite & Mather (2008)* and *Amor et al. (2016)* for *O. insularis* in Brazil (Figs. 2–5; Table 5). The few differences found, as the smaller eSDI, the smaller MWI or the smaller WDI, could be attributed to local adaptation (*Guerra et al., 2010*) or, perhaps, to slight tissue deformations derived from the fixation and preservation process (*Allcock et al., 2011*). In fact, SIMPER analysis revealed that differences between the VRS common octopus and *O. insularis* specimens from Brazil were mainly attributed to traits related to the shape of the web and mantle (e.g. WDI and MWI), which are more likely to suffer from fixation and preservation artefacts.

Moreover, *Amor et al. (2016)* investigated the morphological relationships among seven phylogenetic clades of the 'Octopus vulgaris species complex' and found several significant morphological differences among sampling localities of conspecifics, considering them to represent population-level differences. Specimens analysed in the

**Table 3 Comparison of morphological traits between octopus taxa.**

**Group VRS common octopus & *Octopus vulgaris sensu stricto***

**Average squared distance = 37.45**

| Trait | Average squared distance | Contribution % | Cumulative % |
| --- | --- | --- | --- |
| HASC | 4.99 | 13.33 | 13.33 |
| TOLI | 4.97 | 13.27 | 26.60 |
| eSDI | 4.8 | 12.82 | 39.42 |
| WDI | 4.63 | 12.37 | 51.79 |
| HWI | 4.3 | 11.49 | 63.28 |
| HcAI | 3.97 | 10.60 | 73.89 |

**Group VRS common octopus & *Octopus insularis* from Brazil**

**Average squared distance = 15.50**

| Trait | Average squared distance | Contribution % | Cumulative % |
| --- | --- | --- | --- |
| WDI | 3.17 | 20.47 | 20.47 |
| MWI | 2.62 | 16.90 | 37.37 |
| CLI | 2.45 | 15.80 | 53.16 |
| HWI | 2.14 | 13.77 | 66.94 |
| LLI | 2.03 | 13.08 | 80.02 |

**Group *Octopus insularis* from Brazil & *Octopus vulgaris sensu stricto***

**Average squared distance = 24.42**

| Trait | Average squared distance | Contribution % | Cumulative % |
| --- | --- | --- | --- |
| HcAI | 4.75 | 19.45 | 19.45 |
| eSDI | 3.99 | 16.34 | 35.79 |
| HASC | 3.55 | 14.54 | 50.33 |
| CLI | 2.72 | 11.15 | 61.49 |
| FLI | 2.62 | 10.69 | 72.17 |
| LLI | 2.19 | 8.95 | 81.12 |

Notes:
Contribution of morphological traits to the average squared Euclidean distance between the VRS common octopus, *O. insularis* from Brazil and *O. vulgaris s. s.* (see Methods for abbreviations of morphological traits).
Morphological traits are listed in decreasing order of Contribution %. Cumulative % does not reach 100% in order to facilitate interpretation.

present study are close to the maximum dimensions reported in Brazil: two kg TW, 700 mm TL and 190 mm ML (*Lima et al., 2017*). Colour patterns and skin texture observed in our specimens exactly match what has been previously reported for *O. insularis*. This species shares the 'patch-and-groove' topology with several other *Octopus* spp. (*Norman, Finn & Hochberg, 2016*); however, the observation of specific colour patterns (e.g. the red/white reticulate skin pattern observed in the inner part of the arms when the octopuses were hidden in the den as well as the alternating light/dark bars and the blue–green ring around the eye; Fig. 5) allows a rapid identification of the species (*Leite et al., 2008*; *Leite & Mather, 2008*).

Our morphological analysis clearly differentiated the VRS common octopus from *O. vulgaris s. s.*, mainly based on sexual traits such as HASC, TOLI and eSDI

**Table 4  Genetic distances.**

| Taxa | Veracruz Reef System common octopus | | | | Reference |
| | COI | COIII | r16S | Rhodopsin | |
|---|---|---|---|---|---|
| *Octopus insularis* VRS | 0.0–0.4 | 0 | 0.0–0.3 | 0 | This study |
| *Octopus insularis* Isla Mujeres, MX | 0.0–0.2 (KX611855) | NA | NA | NA | *Lima et al. (2017)* |
| *Octopus insularis* Brazil | 0.0–0.2 (KF843999–KF844025) | 0 (AJ012123) | 0.0–0.6 (KF843956–KF843969) | NA | *Sales et al. (2013) Warnke et al. (2004)* |
| *Octopus maya* Mexico | 0.8–0.9 (KX611862, KX611863) | 5.6 (KX219650) | 3.8–3.9 (KX219653) | NA | *Flores-Valle et al. (2018) Lima et al. (2017)* |
| *Octopus mimus* Chile /MX | 0.6 (GU355926) | 5.3 (AJ012128) | 4.7–5.1 (AJ390318) | 0.5 (KT335848) | *Acosta-Jofré et al. (2012) Warnke et al. (2004)* |
| *Octopus bimaculatus* MX | 11.2–11.5 (KT335828) | 5.5 (X83100) | 6.3 (KT335834) | 1.7 (KT335846) | *Flores-Valle et al. (2018) Barriga-Sosa et al. (1995)* |
| *Octopus bimaculoides* MX | 11.1–11.4 (KF225006) | 6.3 (KF225012) | 6.9 (KF373765) | 1.7 (KT335847) | *Pliego-Cárdenas et al. (2014)* |
| *Octopus vulgaris* Gulf of Mexico | 12.1–12.3 (KX611852–KX611854) | NA | NA | NA | *Lima et al. (2017)* |
| *Octopus vulgaris s. s.* France | 12.8–13.1 (EF016328) | 12.6 (AJ012121) | 6.9–7.0 (EF016336) | NA | *Allcock et al. (2006) Warnke et al. (2004)* |
| *Octopus vulgaris* Brazil | 11.8–12.1 (KF844026–KF844041) | 7.7 (AJ616312) | 7.1–7.7 (KF843970–KF843991) | NA | *Sales et al. (2013) Warnke et al. (2004)* |
| *Octopus vulgaris* Japan | 12.7–13.0 (AJ616307) | 10.5 (AJ616311) | 7.3–7.4 (AB430546) | NA | *Kaneko, Kubodera & Iguchis (2011) Warnke et al. (2004)* |
| *Octopus vulgaris* South Africa | 12.8–13.1 (HM104262) | 12.6 (AJ628241) | 6.6 (DQ683228) | 2.4 (HM104297) | *Guzik, Norman & Crozier (2005) Strugnell et al. (2014) Teske et al. (2007)* |
| *Octopus tetricus* Australia | 12.4–12.6 (KJ605251) | 11.5 (AJ628237) | 7.2–7.3 (KJ605231) | NA | *Amor et al. (2014) Guzik, Norman & Crozier (2005)* |
| *Octopus cyanea* Japan | 15.0–15.3 (AB191280) | 14.1 (AJ628220) | 9.2–9.3 (GQ900721) | NA | *Guzik, Norman & Crozier (2005) Huffard et al. (2010) Takumiya et al. (2005)* |
| *Octopus hummelincki* Brazil | 14.2–14.4 (KF844044) | NA | 11.8 (KF843996) | NA | *Sales et al. (2013)* |
| *Eledone massyae* Brazil | 18.9–19.2 (KF844046) | NA | 15.4 (KF843998) | NA | *Sales et al. (2013)* |
| *Cistopus indicus* West Pacific Ocean | 17.8 (HM104258) | 18.3 (AJ628208) | 12.5 (AJ252744) | 3.8 (HM104291) | *Guzik, Norman & Crozier (2005) Strugnell et al. (2014)* |
| *Amphioctopus* sp. Brazil | 18.1–18.7 (KF844045) | NA | 7.9 (KF843997) | NA | *Sales et al. (2013)* |

Notes:
Tamura–Nei average genetic distances (%) between the specimens from the Veracruz Reef System and related octopus species for COI, COIII, r16S and rhodopsin gene regions. GenBank accession numbers are provided for comparison.
NA, Not available; MX, Mexico.

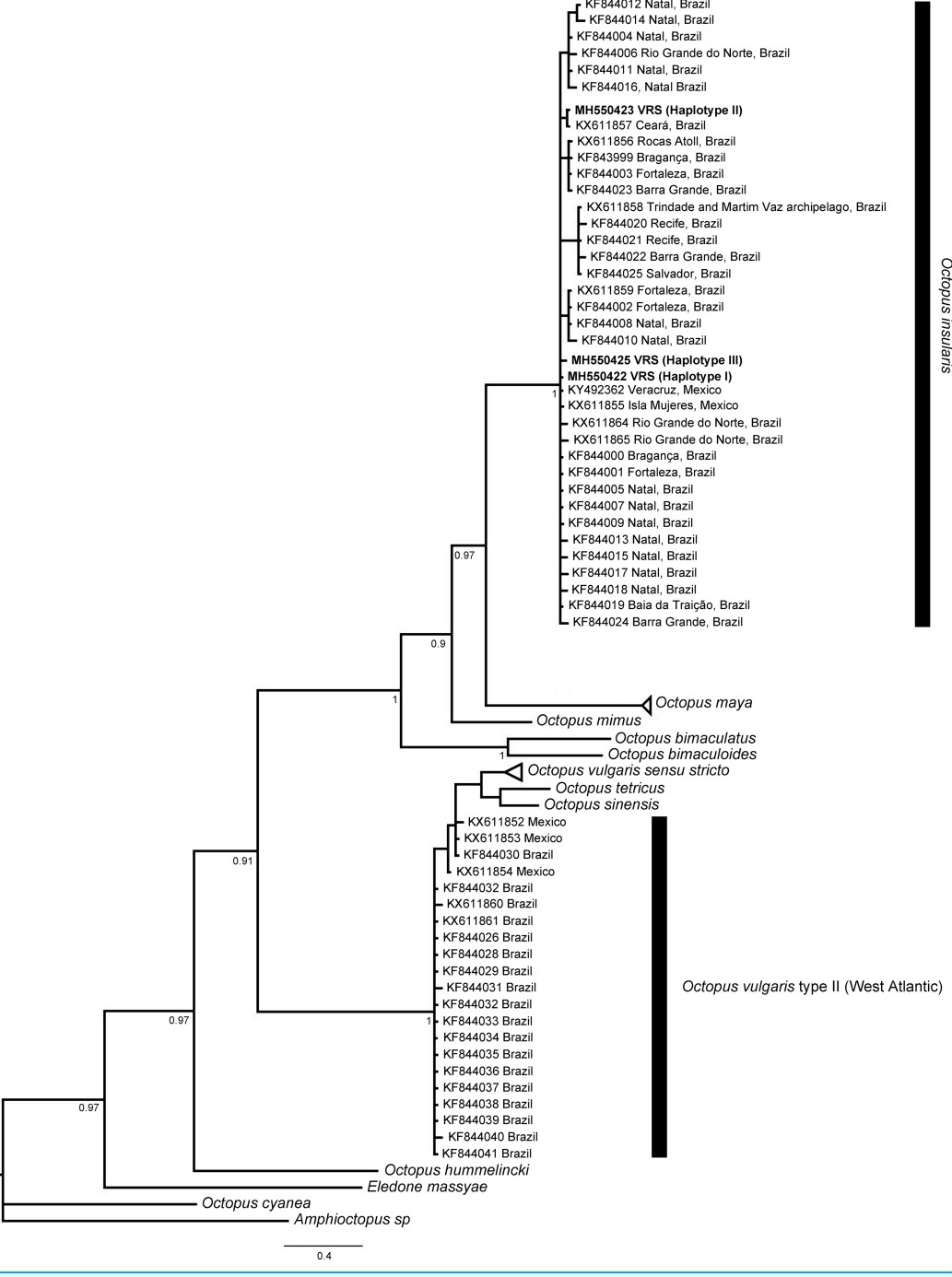

**Figure 7 Bayesian phylogenetic tree based on COI sequences.** Shows the VRS common octopus (*O. insularis*) clade and the *O. vulgaris* type II clade. Each node is labeled with its posterior probability.

(Fig. 6; Table 3). These results support the observations of *Amor et al. (2016)*, whom report that the main morphological differences among members of the *O. vulgaris* species complex were driven by male sexual traits. Moreover, our morphological data on the VRS common octopus strongly differ from the data reported for *O. vulgaris s. s.* elsewhere

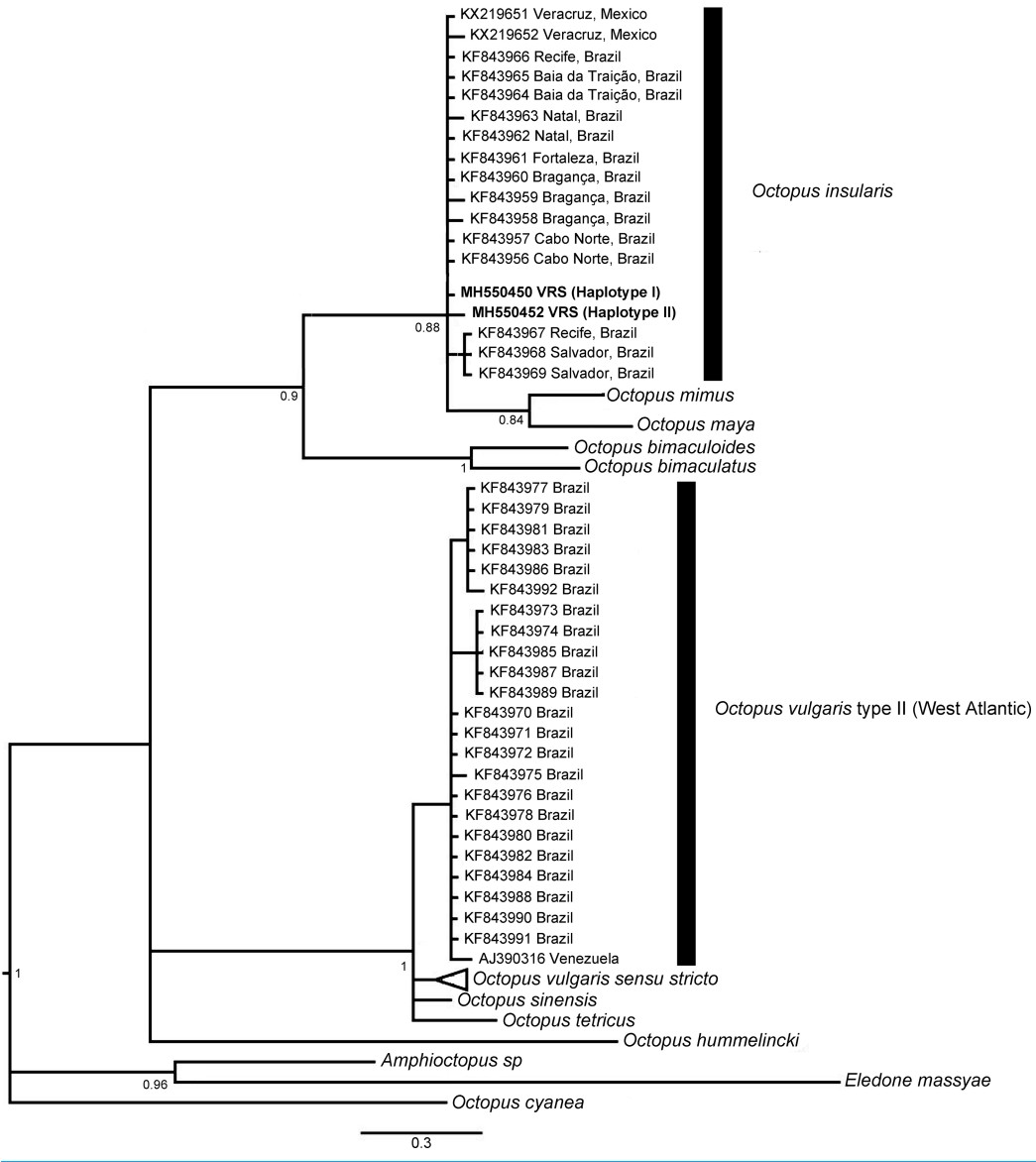

**Figure 8 Bayesian phylogenetic tree based on r16S sequences.** Shows the VRS common octopus (*Octopus insularis*) clade and the *O. vulgaris* type II clade. Each node is labeled with its posterior probability.

(*Mangold, 1998*; *Otero et al., 2007*; *Amor et al., 2016*) (Table 5). The VRS common octopus has a smaller size (189 mm vs. 350 mm max ML), fewer suckers in the hectocotylized arm (HASC 103–146 vs. 156–183), smaller enlarged suckers (eSDI 8.87–13.75 vs. 16.67–25.60), smaller calamus (CLI 40.79–58.56 vs. 40.39–67.55), larger ligula (LLI 0.92–1.65 vs. 0.66–1.29), shallower web (WDI 16.35–24.91 vs. 82.09–146.63) and smaller spermatophores (SpLI 28.06–38.68 vs. 31.00–81.00). Another notable difference between both species is the absence of enlarged suckers in *O. insularis* females while they are present in *O. vulgaris* (*Mangold, 1998*; *Norman, Finn & Hochberg, 2016*).

The common octopus of the VRS can also be differentiated from similar taxa known to inhabit the western Atlantic based on several morphological characters. In this sense,

Table 5 Morphological comparison of the VRS common octopus with similar taxa.

| Parameter | VRS common octopus | | O. insularis | | O. vulgaris s. s. | | O. tayrona | | O. maya | |
|---|---|---|---|---|---|---|---|---|---|---|
| | This study | | Leite et al. (2008), Amor et al. (2016) and Lima et al. (2017) | | Mangold (1998), Otero et al. (2007) and Amor et al. (2016) | | Guerrero-Kommritz & Camelo-Guarin (2016) | | Voss & Solís-Ramírez (1966) and Lima et al. (2017) | |
| | Min | Max | Min | Max | Min | Max | Min | Max | Min | Max |
| Mantle length | 101 | 189 | 80 | 190 | 80 | **350** | **24** | **130** | 48 | 210 |
| Head width index | 27.67 | 50.13 | 35.00 | 48.00 | **43.58** | **61.90** | 29.20 | 104.16 | 27.00 | 48.00 |
| Calamus length index | 40.79 | 58.56 | 41.00 | 56.00 | **40.39** | **67.55** | **20.00** | **50.00** | 24.00 | 27.00 |
| Ligula length index | 0.92 | 1.65 | 1.30 | 1.70 | **0.66** | **1.29** | 0.42 | 1.62 | 1.40 | 1.90 |
| Enlarged sucker diameter index | 8.87 | 13.75 | 9.19 | 16.00 | **16.67** | **25.60** | – | – | – | – |
| Hectocotylized arm index | 229.56 | 306.88 | 188.87 | 320.44 | **320.18** | **528.85** | 96.85 | 558.30 | 216 | 348 |
| Hectocotylized arm sucker count | 103 | 146 | 96 | 142 | **156** | **183** | 112 | 135 | – | – |
| Funnel length index | 26.79 | 49.47 | 28.95 | 49.00 | 18.79 | 52.52 | 11.90 | 79.20 | – | – |
| Mantle width index | 45.85 | 80.21 | 59.00 | 95.00 | 63.56 | 83.14 | **45.20** | **112.50** | 42.00 | 64.00 |
| Web depth index | 16.35 | 24.91 | 22.00 | 29.00 | **82.09** | **146.63** | **8.90** | **82.40** | 16.00 | 30.00 |
| Spermatophore length index | 28.06 | 38.68 | 27.00 | 43.00 | **31.00** | **81.00** | – | – | 47.00 | 60.00 |
| Gill count | 8–11 | | 8–11 | | 9–10 | | 9–12 | | 9–10 | |
| Ocelli | Absent | | Absent | | Absent | | Absent | | **Present** | |
| Post-hatching lifestyle | Merobenthic | | Merobenthic | | Merobenthic | | Merobenthic | | **Holobenthic** | |

Note:
Main differences are shown in bold.

*O. insularis* from Veracruz can be distinguished from *O. tayrona* from the Colombian Caribbean based on the presence of enlarged suckers, larger size of mature specimens (189 mm vs. 130 mm max ML), larger calamus (CLI 40.79–58.56 vs. 20.00–50.00), narrower mantle (max MWI 80.21 vs. 112.50) and shallower web (max WDI 24.91 vs. 82.40) (Table 5). *Octopus insularis* and *O. maya* Voss & Solís, 1966, an abundant species endemic to the Campeche Bank, southeastern Gulf of Mexico, are genetically considered sister species (*Sales et al., 2013*). However, the latter is immediately identified by the presence of a dark ocellus below each eye, and its large eggs (*Voss & Solís-Ramírez, 1966*). *Octopus briareus* Robson, 1929 is a smaller species (120 mm max ML) and has a larger ligula (LLI 3–4), smaller calamus (CLI 28–32), fewer gill lamellae (6–8) and a distinct iridescent blue–green colour in life (*Voss & Toll, 1998*). *Octopus hummelincki*, a common reef-associated octopus, is smaller (72 mm max ML), and possesses a larger ligula (LLI 3–5), fewer gill lamellae (5–9) and a pair of ocelli consisting of a dark central spot inside a conspicuous iridescent blue ring (*Voss & Toll, 1998*). Lastly, the artisanal fishermen of the VRS sometimes manage to capture specimens of the locally known as 'pulpo malario', which so far is thought to be *Callistoctopus macropus* (Risso, 1826).

However, in light of its original description from the Mediterranean Sea, a critical revision has been suggested for this taxa in the western Atlantic (*Leite et al., 2008*). The species can be easily differentiated from *O. insularis* by its brick red colour with distinct pattern of white spots on dorsal mantle, head and arms as well as by its larger ligula, longer arms, shallower web and very reduced stylets (*Mangold, 1998*).

The resolved COI and r16S highly supported clades, one including the monophyletic clade, which we refer to as the American *Octopus* clade, conformed by *O. bimaculatus*, *O. bimaculoides*, *O. insularis*, *O. maya* and *O. mimus*, along with the specimens from the VRS; and the *O. vulgaris* clade, are concordant results to those that have been previously reported elsewhere (*Leite et al., 2008*; *Sales et al., 2013*; *Lima et al., 2017*; *Flores-Valle et al., 2018*). These latest reports resolved two main and highly supported clades (*O. insularis* and *O. vulgaris* clades).

The genetic similarities found between the specimens analysed from the southern reefs Isla de Enmedio and Anegada de Afuera of the VRS and *O. insularis* from Brazil also support the identity of the formers as *O. insularis*, as they share haplotypes in all mitochondrial genes analysed (e.g. average genetic distance 0.0–0.6%; see Table 4). Most samples from the VRS shared r16S Haplotype 1 with *O. insularis* from the northern coast of Brazil (*Sales et al., 2013*); COI Haplotype 1 is also shared with *O. insularis* from the Brazilian coast and the Fernando de Noronha archipelago (*Sales et al., 2013*; *Lima et al., 2017*), whereas Haplotype 2 is shared with *O. insularis* from the São Pedro and São Paulo archipelago (*Lima et al., 2017*). The only Haplotype resolved by COIII, is shared with haplotype AJ012123, from Brazil (*Warnke et al., 2004*). Unfortunately, the lack of available rhodopsin sequences of *O. insularis* from Brazil in GenBank precluded a comparison with the specimens from the VRS. However, the nuclear genetic distance between *O. vulgaris* and *O. insularis* was the highest among congeners (2.4% average genetic distance). This result supports the distinction of VRS specimens from *O. vulgaris*.

In this study, we proved, based on an integrative taxonomic approach, that the common octopus that supports the main cephalopod fishery of the southwestern Gulf of Mexico is *O. insularis*. This fact is consistent with the first record of this species in the Gulf of Mexico by *Flores-Valle et al. (2018)*. These authors reckon, however, the need for a detailed morphological description to demonstrate that the Mexican and Brazilian taxa are conspecifics. This matter has been fully resolved in the present study by combining both morphological and genetic analyses.

In the light of our findings, we infer that previous published data considering *O. vulgaris* as the common octopus of the VRS (e.g. *Jiménez-Badillo & Castro-Gaspar, 2007*; *Jiménez-Badillo, 2010*; *Jiménez-Badillo, 2013*) should in fact be attributed to *O. insularis*. It has been suggested that *O. insularis* and *O. vulgaris*, although in sympatry, might be occupying different niches related to depth and temperature in northeastern Brazil, with the former inhabiting shallower and warmer waters (*Lima et al., 2017*). The reason for this difference seems to be the higher tolerance of *O. insularis* to both salinity increases and decreases, as evidenced by osmotic experiments (*Amado et al., 2015*). This explanation is consistent with the presence of *O. insularis* in estuaries of small rivers and in tide pools in Brazil, where salinity and temperature can vary greatly (e.g. 36–42 PSU and 24–36 °C)

(*Fonseca, Villaça & Knoppers, 2012*; *LIMA, 2017*) and with its occurrence in the shallow waters of the VRS, where significant changes in salinity (e.g. from 32 to 39 PSU) and temperature (e.g. from 19.6 to 30 °C) can occur as a consequence of high evaporation or local rivers discharge, especially under the influence of strong winds (*Salas-Monreal et al., 2009*; *Avendaño-Alvarez et al., 2017*).

The Caribbean Sea has recently been suggested by *LIMA (2017)* as an origin area of *O. insularis*, which presumably diverged from other *Octopus* spp. after the uplift of the Panama Isthmus. The fact that *O. insularis* is commonly found within the VRS in shallow waters along the coast and on many reef lagoons, supports the hypothesis of a wide distribution of the species linked to a high dispersal potential, including the shallow waters of the continental shelves, banks, seamounts and islands, in the western Atlantic Ocean (*Leite et al., 2008*; *Lima et al., 2017*). The VRS constitutes, up to now, the north-western limit of a well-established *O. insularis*' population, however, additional sampling within the Gulf of Mexico and other areas along the western Atlantic coast could expand its geographical dominance in tropical waters and include, for example the Lobos-Tuxpan Reef System, the Alacranes Reef System, or the Mesoamerican Reef System. Indeed, a priori in situ identifications based on colouration patterns (see Fig. 9) point to the presence of the species in the coral reef system of Puerto Morelos, Mexico, just a few km south of Isla Mujeres, where another specimen was morphologically identified in the field as *O. insularis* (*Lima et al., 2017*). Nevertheless, proving the existence of a population there would require a formal analysis of octopus specimens across the area to determine genetic cohesion.

Recognizing *O. insularis* as the primary octopod targeted by the shallow-water fishery in the state of Veracruz has implications regarding the taxonomic composition of Mexican octopus fishery data. Until *Voss & Solís-Ramírez's (1966)* description of *O. maya*, a large size holobenthic octopus endemic to the shallow waters of the Yucatan peninsula, all similar-sized octopuses captured in the Mexican Atlantic were considered as *O. vulgaris*. As a result of the significant dominance of *O. maya* in commercial landings, management policies for the Mexican Atlantic octopus fishery have been based on its biology since the 80's (*Diario Oficial de la Federación (DOF), 2012*, *2014*). In spite of the existence of a separate fishery at the VRS, its peculiarities have only been recently recognized, with the establishment of separate management measures such as different fishing gears and closures (*Diario Oficial de la Federación (DOF), 2016*). Differentiation between *O. vulgaris* and *O. maya* was somewhat easier that the one concerning *O. insularis* because *O. maya* does not have paralarval stage and lays fewer but much larger eggs (*Voss & Solís-Ramírez, 1966*). The superficial similarities between *O. vulgaris* and *O. insularis* posed more difficulties assessing the taxonomic identity of the latter species and made it necessary to conduct detailed morphological and genetic analyses in order to differentiate them. Consequently, minding that *O. insularis* is the main species captured in the southwestern Gulf of Mexico, we suggest that it should be included in the statistics as being responsible for a significant amount of the total catch taken by Mexican fishers and reported through FAO as *O.* 'vulgaris' type I (*Food and Agriculture Organization of the United Nations (FAO), 2016*; *Norman, Finn & Hochberg, 2016*). Moreover, the most recent

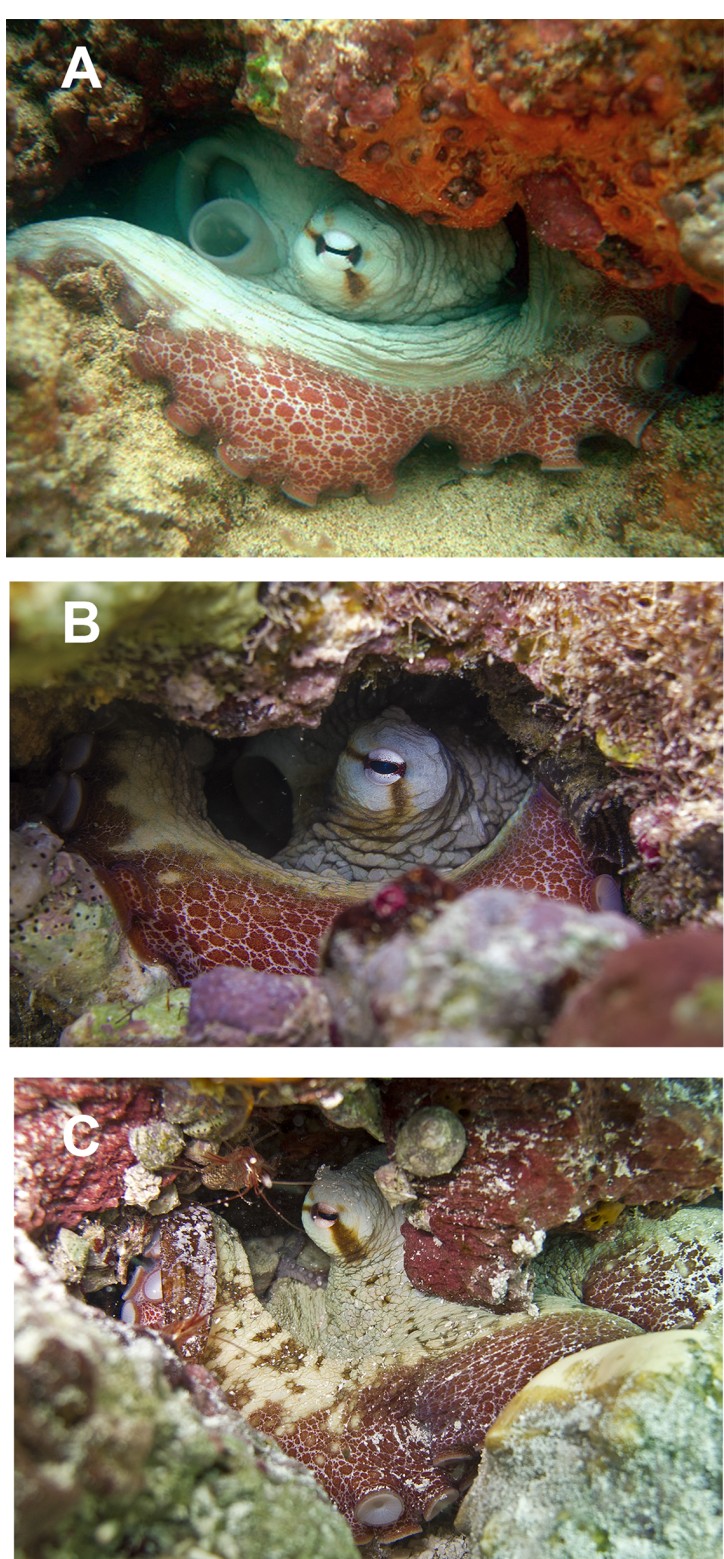

**Figure 9 In situ photographs of octopus specimens.** (A) *Octopus insularis* from Brazil; (B) *O. insularis* from Veracruz, Mexico. (C) *O.* cf. *insularis* from Puerto Morelos Reef National Park, Quintana Roo, Mexico (Photo credits A: Tatiana S. Leite; B, C: Roberto González-Gómez).

studies dealing with *O. vulgaris* type I identifications have shown that the specimens had been misidentified in all cases, actually grouping in the same clade as *O. insularis*, *O. maya*, or *O. vulgaris* type II from Brazil (*Lima et al., 2017*; *Flores-Valle et al., 2018*; this study); therefore, we cast doubt on the utility of this taxon. In accordance, Mexican management plans concerning the common octopus of the VRS (*Diario Oficial de la Federación (DOF), 2012*, *2014*, *2016*) should be readdressed to include *O. insularis* as the targeted species, to achieve more accurate fishery statistics and avoid critical population changes going unnoticed.

Misidentifications are common among different commercially exploited octopus species and are thought to occur due to a lack of knowledge about useful diagnostic characters (*Lima et al., 2017*). As these authors suggest, identification of specimens should occur immediately after capture, because it is easier to recognize distinct morphological characters in fresh specimens. In line with this, we believe that fishermen and warehouse owners represent an important sector that could make a difference towards successful management plans derived from proper octopus identification. Hence, the distribution of a visual identification guide of the VRS octopus species (currently in preparation) including colour photographs of live and dead specimens as well as key characters of each species could aid to achieve this important goal.

## CONCLUSIONS

Proper identification of organisms is necessary to achieve accurate estimates of biodiversity and is particularly important in commercially exploited species, because it allows the effective management of their stocks. The VRS common octopus has been mistaken with *O. vulgaris* until now due to superficial morphological similarities between both taxa. In this study, following an integrative taxonomic approach, we provide morphological and genetic evidence for the identity of the former as *O. insularis*. Morphological analyses were successful in distinguishing both taxa, with main differences based on male sexual traits such as the number of suckers in the hectocotylized arm or the diameter of enlarged suckers. Hence, our study shows a new case of misidentification involving *O. vulgaris* and highlights the need of more morphological and genetic studies regarding the species of the 'Octopus vulgaris complex' in the western Atlantic in order to properly address the management of tropical octopus fisheries and their ecological implications.

## ABBREVIATIONS OF MEASUREMENTS AND INDICES

**TW**      total wet weight
**TL**      total length
**ML**      dorsal mantle length
**MWI**      mantle width index (mantle width/ML $\times$ 100)
**MAI**      mantle arm index (ML/longest arm length $\times$ 100)
**HWI**      head width index (head width/ML $\times$ 100)
**AL**      arm length (of intact arms, measured from mouth to the tip of the arm over the row of suckers)
**ALI**      arm length index (arm length/TL $\times$ 100)

| AW | arm width |
| --- | --- |
| AWI | arm width index (arm width at the widest point of the stoutest arm/ML $\times$ 100) |
| ASC | arm sucker count |
| HASC | hectocotylized arm sucker count |
| GiLC | gill lamellae count (number of outer gill lamellae including the terminal lamella) |
| FLI | funnel length index (funnel length/ML $\times$ 100) |
| HAL | hectocotylized arm length |
| HcAI | hectocotylized arm index (HAL/ML $\times$ 100) |
| OAI | opposite arm index (length of hectocotylized arm as a percentage of its fellow arm on opposite side) |
| LL | ligula length |
| LLI | ligula length index (LL/HAL $\times$ 100) |
| CL | calamus length |
| CLI | calamus length index (CL/LL $\times$ 100) |
| nSD | normal sucker diameter |
| nSDI | normal sucker diameter index (largest nSD/ML $\times$ 100) |
| eSD | enlarged sucker diameter |
| eSDI | enlarged sucker diameter index (largest eSD/ML $\times$ 100) |
| ELD | eye lens diameter |
| EDI | eye lens diameter index (ELD/ML $\times$ 100) |
| WD | web depth |
| WDI | web depth index (WD/ML $\times$ 100) |
| TOL | terminal organ length |
| TOLI | terminal organ length index (TOL/ML $\times$ 100) |
| SpL | spermatophores length |
| SpLI | spermatophore length index (length of spermatophore/ML $\times$ 100). |

## ACKNOWLEDGEMENTS

This work was made possible by the invaluable help of the artisanal fishermen from the 'Cooperativa Arrecifes de Antón Lizardo.' The authors wish to express their gratitude to all of them for their hospitality and good company during the sampling trips. Christine Huffard kindly reviewed and improved a preliminary draft of this paper by providing many helpful comments. We thank Paula Rothman for her great effort helping us getting very helpful literature.

### Funding

Financial support was provided to Roberto González Gómez by the PADI Foundation and by the CONACYT PhD scholarship 464700. The study was also supported by grant No. UAMI-14709004 to Irene de los Angeles Barriga Sosa. There was no additional

external funding received for this study. The funders had no role in study design, data collection and analysis, decision to publish, or preparation of the manuscript.

## Grant Disclosures
The following grant information was disclosed by the authors:
PADI Foundation.
CONACYT PhD scholarship 464700.
UAMI-14709004.

## Competing Interests
The authors declare that they have no competing interests.

## Author Contributions

- Roberto González-Gómez conceived and designed the experiments, performed the experiments, analysed the data, contributed reagents/materials/analysis tools, prepared figures and/or tables, authored or reviewed drafts of the paper, approved the final draft.
- Irene de los Angeles Barriga-Sosa conceived and designed the experiments, performed the experiments, analysed the data, contributed reagents/materials/analysis tools, prepared figures and/or tables, authored or reviewed drafts of the paper, approved the final draft.
- Ricardo Pliego-Cárdenas performed the experiments, analysed the data, prepared figures and/or tables, authored or reviewed drafts of the paper, approved the final draft.
- Lourdes Jiménez-Badillo conceived and designed the experiments, contributed reagents/materials/analysis tools, authored or reviewed drafts of the paper, approved the final draft.
- Unai Markaida conceived and designed the experiments, contributed reagents/materials/analysis tools, authored or reviewed drafts of the paper, approved the final draft.
- César Meiners-Mandujano conceived and designed the experiments, performed the experiments, analysed the data, contributed reagents/materials/analysis tools, authored or reviewed drafts of the paper, approved the final draft.
- Piedad S. Morillo-Velarde conceived and designed the experiments, performed the experiments, contributed reagents/materials/analysis tools, authored or reviewed drafts of the paper, approved the final draft.

## DNA Deposition
The following information was supplied regarding the deposition of DNA sequences:
New-generated sequences are provided as Supplemental Files and at GenBank: MH550422 to MH550467.

## Data Availability
Raw data is provided in the Supplemental Files.

## Supplemental Information
Supplemental information for this article can be found online at http://dx.doi.org/10.7717/peerj.6015#supplemental-information.

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
