# Peer review of "An integrative taxonomic approach reveals Octopus insularis as the dominant species in the Veracruz Reef System (southwestern Gulf of Mexico)"

_PeerJ, doi:10.7717/peerj.6015_

## Round 0.1 · original submission · Minor Revisions

As you will see, the two expert reviewers reacted very positively to your manuscript (as do I) and suggested a few relatively minor modifications. I agree that showing a PCO plot would be very useful.

·

Basic reporting

This is a clear well-written paper that tackles an important subject and provides a definitive answer. Most of the figures and tables are necessary although I would suggest figure 5 and figure 9 could be merged. The text size on figures 7 & 8 needs to be increased. Although abbreviations are fine for localities in figures and tables, I think it would be helpful if VRS was written in full throughout the text. Genbank accession numbers are reported for new sequences.
Some points of grammar:
Line 68: ‘lies’ not ‘lays’
Line 73: ‘to date’ not ‘up to date’
Line 90: “it is now known” not “now it is known”
Line 208: ‘data are’ not ‘data is’
Line 324: “in contrast’ not ‘On the other hand’. In English, ‘on the other hand’ is very informal.
Line 324: “differentiating” not “differencing”
Line 331: “shared” not “share”
Line 378: delete “on the other hand” it is not necessary

Experimental design

This is original primary research which falls within the aims and scope of the journal. The question is an important one concerning correct identification of a commercially fished species. The paper uses standard techniques which are well executed and I find little to criticize within the methods. They are well described and the data are clearly presented.
Authors should check that they are always citing the paper they intended and citing correctly. For example, Line 209, I think the authors are referring to Allcock, Strugnell & Johnson, 2008. Similarly Line 215.
Authors might also check references in Table 4. I think, for example, that Allcock et al. 2006 included O. vulgaris from France, not from Brazil. Sequences from Cistopus indicus are available in both Gukiz et al. 2005 and Strugnell et al. 2014, and additional rhodopsin sequences may be found in Strugnell et al. 2014.

Validity of the findings

This is all well done and I find little to criticize. Perhaps the authors could add a little more on the identification of O. insularis versus O. vulgaris based on the external skin sculpture. Norman and Hochberg have often mentioned the classic "patch and groove" sculpture of Octopus s.s., so I think it would be helpful to include a more directly stated comparison of how O. insularis external colour/sculpture differens from "patch and groove"

·

Basic reporting

It is a well structured and integrative study dealing with a complex issue such as redefining a "plastic" octopod species originally identified as Octopus vulgaris.

The authors did a great morphological and genetic job to show that the species inhabiting the Veracruz reef system is O. insularis, rather than O. vulgaris.

I only have minor comments that are attached to the pdf.

1. I would like to see a PCO plot rather than MDS because it shows the true differences in mulditimensional space for the 10 traits measured in two axis that account for a certain % of total variance, rather than maximizing these differences in the first two axis which is the representation of MDS.

2. I would suggest to concatenate the sequences obtained to have a better phylogenetic resolution.

Experimental design

Good experimental design, enough samples analysed for the morphological and genetic analyses.

The authors need to clarify why the number of specimens analysed for genetics (n=24, line 224) are different than those specified on line 324 (n=19). Indeed, the number of sequences deposited on genbank N= 46 (MH550422-MH550467) is less than the number os genes analyzed (4) and the specimens indicated on line 324 (19x4=76).

Also, I missed a table showing the details of the sequences retrieved form Genbank (Acc number, location, species, authors...) so other authors can have a clear picture of the comparisons carried in this study.

Validity of the findings

Great job done by the authors, with important findings that will hopefully change management strategies off the coast of Mexico and Caribean Sea.

Conclusion is well stated and linked to the origial research question.

Additional comments

Overall, a good work with minor things to be changed. Great effort and a good morphological and genetic analysis.

---

## Round 0.2 · accepted · Accept

Thank you for addressing the suggested "minor revisions" so thoroughly. I am happy to accept your manuscript at this time.

#